# Lipoprotein (a) and the Occurrence of Lipid Disorders and Other Cardiovascular Risk Factors in Patients without Diagnosed Cardiovascular Disease

**DOI:** 10.3390/jcm13164649

**Published:** 2024-08-08

**Authors:** Jakub Ratajczak, Aldona Kubica, Łukasz Pietrzykowski, Piotr Michalski, Agata Kosobucka-Ozdoba, Krzysztof Buczkowski, Magdalena Krintus, Piotr Jankowski, Jacek Kubica

**Affiliations:** 1Department of Cardiac Rehabilitation and Health Promotion, Collegium Medicum in Bydgoszcz, Nicolaus Copernicus University in Torun, 85-094 Bydgoszcz, Poland; jakub.ratajczak@cm.umk.pl (J.R.); akubica@cm.umk.pl (A.K.); piotr.michalski@cm.umk.pl (P.M.); a.kosobucka@cm.umk.pl (A.K.-O.); 2Department of Family Medicine, Collegium Medicum in Bydgoszcz, Nicolaus Copernicus University in Torun, 85-094 Bydgoszcz, Poland; buczkowskik@cm.umk.pl; 3Department of Laboratory Medicine, Collegium Medicum in Bydgoszcz, Nicolaus Copernicus University in Torun, 85-094 Bydgoszcz, Poland; krintus@cm.umk.pl; 4Department of Internal Medicine and Geriatric Cardiology, Center of Postgraduate Medical Education, 01-813 Warsaw, Poland; piotrjankowski@interia.pl; 5Department of Epidemiology and Health Promotion, School of Public Health, Center of Postgraduate Medical Education, 01-826 Warsaw, Poland; 6Department of Cardiology and Internal Medicine, Collegium Medicum in Bydgoszcz, Nicolaus Copernicus University in Torun, 85-094 Bydgoszcz, Poland; jkubica@cm.umk.pl

**Keywords:** lipoprotein (a), Lp(a), cardiovascular risk factors, dyslipidemia

## Abstract

**Background**: Elevated lipoprotein (a) [Lp(a)] concentrations are linked mainly to genetic factors. The relationship between Lp(a) and other lipid disorders or cardiovascular (CV) risk factors has been less investigated. The aim of this study was to assess the occurrence of lipid disorders and other CV risk factors according to Lp(a) concentrations. **Methods**: A cross-sectional analysis of 200 primary-care patients who had not been diagnosed with CV disease was conducted. The following risk factors were assessed: older age, history of hypertension, diabetes mellitus or dyslipidemia, smoking, lack of physical activity, body mass index (BMI), and waist circumference. The following lipid parameters were measured: total cholesterol (TC), low-density lipoprotein cholesterol (LDL-C), non-high-density lipoprotein cholesterol (non-HDL-C), high-density lipoprotein cholesterol (HDL-C), triglycerides (TG), and small, dense LDL (sdLDL-C). Patients were divided into two groups based on their Lp(a) concentrations: <30 mg/dL and ≥30 mg/dL. **Results**: In 70% of patients, the Lp(a) concentration was <30 mg/dL. The concentrations of lipid parameters did not differ between the groups. The rate of patients with sdLDL-C >1.0 mmol/L was higher in the low-Lp(a) group (10.0 vs. 1.7%, *p* = 0.04), with no significant differences regarding the other analyzed lipid disorders (*p* > 0.05). Both in the low- and high-Lp(a) group, most patients had two other abnormal lipid factors (45.0% and 60.0%, respectively). The distribution of impaired lipid parameters (*p* = 0.41) and other CV risk factors (*p* = 0.16) was similar in both groups. There was a lower rate of patients >60 years old (15.0% vs. 32.9%, *p* = 0.01) and with a BMI ≥ 25 kg/m^2^ (46.7% vs. 63.6%, *p* = 0.026) in the high-Lp(a) group, and previously diagnosed hyperlipidemia was more prevalent in this group (65.0% vs. 47.1%, *p* = 0.02). The occurrence of other cardiovascular risk factors did not differ significantly between the Lp(a) groups (*p* > 0.05). In the high-Lp(a) group, the highest proportion (25.0%) had two CV risk factors, and in the low-Lp(a) group, 31.4% had four CV risk factors. **Conclusions**: An elevated Lp(a) concentration is not related to the number of conventional CV risk factors or other impairment major lipid parameters.

## 1. Introduction

Dyslipidemia is one of the major risk factors for coronary atherosclerotic cardiovascular disease (ASCVD) worldwide, including in Poland [1,2]. Previous studies showed unsatisfying control of the main cardiovascular (CV) risk factors in addition to poor management of lipid disorders [3,4,5]. The burden of dyslipidemia in ASCVD is related to the low detection rate and inadequate lipid-lowering treatments even for high-risk patients [1,6]. The adverse impact of low-density lipoprotein cholesterol (LDL-C) on ASCVD development has been confirmed in numerous, repeated studies, and LDL-C concentration remains the major therapeutic target for reducing dyslipidemia-related risk [7,8]. Non-high-density lipoprotein cholesterol (non-HDL-C), apolipoprotein B (apoB), and LDL-C concentrations are highly correlated; thus, the corresponding information about ASCVD risk is similar [7].

However, under specific circumstances [e.g., for patients with very-low LDL-C levels, diabetes mellitus (DM), a high concentration of triglycerides (TG), obesity], assessments based on LDL-C might underestimate the ASCVD risk, and non-HDL-C or direct apoB measurements are superior [7,9]. Therefore, recent guidelines on CV prevention recommend the use of the non-HDL-C assessments in the new Systematic Coronary Risk Estimation 2 (SCORE2) risk chart [10]. Nevertheless, a lipid profile analysis should be an integral part of a patient’s CV risk assessment in both primary and secondary prevention [10,11]. The growing body of evidence also confirms the adverse effect of other, less commonly examined particles, e.g., lipoprotein (a) [Lp(a)], on overall CV risk [12,13]. In a large analysis of UK Biobank participants, it was observed that Lp(a) increased the risk of ASCVD by 11% per each 50 nmol/L increment [14]. An elevated Lp(a) concentration was also associated with aortic valve stenosis and peripheral artery disease [13]. Lp(a) concentrations are determined mainly by the genetic factors related to the variations of the LPA gene [15]. Recent guidelines recommend conducting an Lp(a) measurement at least once in a lifetime to assess an individual’s risk of ASCVD, and a high risk is assigned to all patients with Lp(a) concentrations >50 mg/dL [7,13].

The aim of the study was to assess the occurrence of lipid disorders and other CV risk factors according to the Lp(a) concentration in patients without a diagnosed coronary artery disease.

## 2. Materials and Methods

This observational, cross-sectional study was performed in 2018–2019. Patients who met the inclusion criteria (Figure 1) were invited for an interview performed by a qualified nurse or a physician. During the initial visit, fasting blood samples were collected to assess the concentration of high-sensitivity troponin I (hsTnI, ng/L), C-reactive protein (CRP, mg/L), serum creatinine (mg/dL), plasma glucose (mmol/L), and the following lipid parameters: total cholesterol (TC, mmol/L); low-density lipoprotein cholesterol (LDL-C, mmol/L); high-density lipoprotein cholesterol (HDL-C, mmol/L); non-high-density lipoprotein cholesterol (non-HDL-C, mmol/L); triglycerides (TG, mmol/L); small, dense LDL-C (sdLDL-C, mmol/L); and lipoprotein (a) [Lp(a), mg/dL]. The patients were divided into two groups based on their Lp(a) concentrations: low-Lp(a) group (<30 mg/dL) and high-Lp(a) group (≥30 mg/dL). Abnormal lipid parameters were defined as TC > 4.9 mmol/L (>190 mg/dL), non-HDL-C > 3.4 mmol/L (>130 mg/dL), LDL-C > 3.0 mmol/L (>115 mg/dL), TG > 1.7 mmol/L (>150 mg/dL), sdLDL-C > 1.0 mmol/L (>40 mg/dL), HDL-C < 1.0 mmol/L (<40 mg/dL) for males, and <1.2 mmol/L (<45 mg/dL) for females [16]. This study compared different categories of Lp(a) and their associations with other impaired lipid parameters such as TC, non-HDL-C, TG, HDL-C, and sdLDL-C. Since LDL-C and non-HDL-C provide similar information on ASCVD risk, only non-HDL-C was included in the analysis of impaired lipid factors, as recommended by the latest guidelines [10]. The height (cm), weight (kg), body mass index (BMI, kg/m^2^), and abdominal circumference (AC, cm) of patients were measured, and self-reported information regarding smoking status and physical activity was collected. Eight major CV risk factors were assessed for each patient. The analyzed CV risk factors included age > 60 years, overweight and obesity (BMI ≥ 25 kg/m^2^), central overweight and obesity (AC ≥ 94 cm for men and ≥80 cm for women), smoker status, inadequate physical activity (<20 min of intensive physical activity 1–2 times a week), and history of hyperlipidemia, hypertension, and DM. A detailed description of the methodology was provided in a previously published paper [4].

The study protocol was approved by the Ethics Committee of The Nicolaus Copernicus University in Torun, Collegium Medicum in Bydgoszcz (KB 586/2017), and this study was conducted in accordance with the Declaration of Helsinki and Good Clinical Practice principles. All participants signed the informed consent prior to their inclusion in the study.

### Statistical Analysis

Statistical analysis was performed with IBM SPSS Statistical software version 27 (Copyright IBM^©^ Corporation, Armonk, NY, USA), and a two-sided *p*-value < 0.05 was applied for statistical significance. The differences between continuous variables were examined with Mann–Whitney test or Student’s *t*-test as appropriate according to the data distribution. To determine the data distribution, the Shapiro–Wilk test and an analysis of histograms were performed. The majority of the analyzed continuous variables had non-normal distribution. Therefore, in order to increase the clarity of the presented results, all continuous variables are shown as medians with interquartile range (IQR). Categorical variables were presented as absolute values and percentages. The differences between the categorical variables were analyzed using chi-squared test.

## 3. Results

A total of 200 patients were included in the analysis. The median age was 52 years (IQR 43.0–60.0), and the majority were men (66.5%). In 70% of patients, the Lp(a) concentration was <30 mg/dL, and in 22.5%, it was >50 mg/dL. The distribution of Lp(a) concentrations for the studied group is presented in Figure 2.

The occurrence of overweight and obesity both defined by BMI (58.5%) and AC (63%) was common. The median TC and LDL-C concentrations were high (5.56 mmol/L and 3.29 mmol/L, respectively). TC > 4.9 mmol/L was observed in 76%, non-HDL-C > 3.4 mmol/L in 74.0%, and LDL-C > 3.0 mmol/L in 61% of patients. The baseline characteristics of the studied group are presented in Table 1.

Table 2 depicts the rates of patients with impaired lipid parameters in the Lp(a) groups. There were no differences regarding the elevated: TC, non-HDL-C, TG, and low-HDL-C categories (*p* > 0.05). Non-HDL-C > 3.4 mmol/L was observed in 72.9% of patients in the low-Lp(a) group and in 73.3% in the high-Lp(a) group. The occurrence of TG > 1.7 mmol/L (>150 mg/dL) was low in both Lp(a) groups, with 20.7% in the low-Lp(a) group and 13.3% in high-Lp(a) group. A higher rate of patients with sdLDL > 1.0 mmol/L (>40 mg/dL) was observed in the low-Lp(a) group (10.0% vs. 1.7%, *p* = 0.04). However, the general prevalence of sdLDL > 1.0 mmol/L was low and occurred in 15 patients (7.5%), out of which only 1 patient was in the high-Lp(a) group.

The median concentrations of lipid parameters were comparable between the Lp(a) < 30 mg/dL and ≥30 mg/dL groups (*p* > 0.05) (Table 3).

Two out of five lipid factors were abnormal for the greatest proportions of patients in both Lp(a) groups (45.0% and 60.0%, respectively). The distribution of the impaired lipid factors (presented in Figure 3) did not differ between the Lp(a) categories (*p* = 0.408).

The most common cardiovascular risk factors in the studied group were a history of hypertension (70.0%), central overweight or obesity (63.0%), and inadequate physical activity (59.5%), as presented in Table 2. There was a lower rate of patients >60 years old in the high-Lp(a) group in comparison to the low-Lp(a) group (15.0% vs. 32.9%, *p* = 0.01). Previously diagnosed hyperlipidemia was more prevalent in the high-Lp(a) group (65.0% vs. 47.1%, *p* = 0.02). Overweight and obesity defined by the BMI was more common among patients with Lp(a) < 30 mg/dL (63.6% vs. 46.7%, *p* = 0.026). The occurrence of other cardiovascular risk factors did not differ significantly between the Lp(a) groups (Table 2).

The distribution of the number of CV risk factors in the Lp(a) groups is presented in Figure 4. The majority of individuals in the low-Lp(a) group exhibited coexistence four (31.4%) or three (20.0%) CV risk factors. In the high-Lp(a) group, most patients had two CV risk factors (25.0%), followed by four CV risk factors (23.35). In this group, the distribution was more diffused, with a less marked peak, in comparison to the low-Lp(a) group. However, the differences regarding the distribution of the number of CV risk factors between the Lp(a) groups did not reach statistical significance (*p* = 0.160).

## 4. Discussion

Lp(a) is an independent factor increasing the risk of CV diseases [17]. The presented study investigated the relationship between Lp(a) concentration and the occurrence of lipid disorders as well as other major conventional CV risk factors. Abnormal lipid parameters occurred with a similar frequency in the Lp(a) groups. A higher rate of patients with sdLDL > 1.0 mmol/L was observed in the low-Lp(a) group. However, this result might be biased by the very low total number of patients with elevated sdLDL levels and should be interpreted with caution. No relationship with the number of lipid disorders was found. The Lp(a) groups did not differ regarding the prevalence of hypertension, DM, tobacco smoking, or physical inactivity. In the group with Lp(a) ≥ 30 mg/dL, lower rates of older patients and individuals with an elevated BMI (≥25.0 kg/m^2^) were observed. This group was also characterized by a higher rate of previously diagnosed hyperlipidemia.

History of hyperlipidemia and the concentration of major lipid parameters were previously analyzed in various studies. Konieczyńska et al. analyzed the middle-aged Polish population without diagnosed ASCVD [18]. In their analysis, the prevalence of dyslipidemia was around 1.7 times greater in the high-Lp(a) group (>100 mg/dL) compared to the moderate (50–100 mg/dL)- and low (<50 mg/dL)-Lp(a) groups, but the prevalence of dyslipidemia in this population was generally low (27.3–46.5%). The authors reported higher levels of TC and LDL-C in the high-Lp(a) groups [18]. It should be mentioned that the population investigated by Konieczyńska et al. was younger (40–65 years of age), and different cut-off points for Lp(a) groups were employed. In a large-population study conducted in the US, the group with Lp(a) > 30 mg/dL had higher levels of TC, LDL-C, and non-HDL-C and lower TG concentrations in comparison to the low-Lp(a) group [19]. The Lp(a) concentration was increased by 6–7% in the group of patients with dyslipidemia due to statin intake [13]. The prevalence of two or more impaired lipid parameters was very high in the presented population (73.5%); however, the distribution of the lipid disorders was similar regardless of the Lp(a) concentration.

The presented analysis shows a lower rate of older patients (≥60 years of age) in a group with elevated Lp(a) concentrations. It is important to note that only patients without diagnosed CVD were included in this study. The concentration of Lp(a) increases early in life and stays at a steady level during adulthood [13]. An elevated Lp(a) concentration has a negative impact on the CV system and increases the risk of myocardial infarction, especially in younger patients [20]. Therefore, the observed lower occurrence of older patients in the high-Lp(a) group may be influenced by the exclusion of patients with established ASCVD.

The association between Lp(a) and DM is an area of interest and requires further investigation. Konieczyńska et al. found no differences between groups with low, moderate, and high Lp(a) concentrations regarding the history of DM [18]. However, the occurrence of DM was generally low (4.7–5.2%) in the investigated population [18]. In contrast, Raitakari et al. found that the prevalence of type 2 DM decreased across the Lp(a) groups, with the highest rate in patients with Lp(a) < 5 mg/dL (5.1%) and the lowest in those with ≥50 mg/dL (1.7%) [21]. On the other hand, in a small analysis of Lp(a) concentrations among patients with DM, patients with insulin-requiring type 2 DM were more likely to have an Lp(a) concentration >30 mg/dL in comparison to type 1 DM patients and healthy controls [22]. Laboudovic et al. presented similar results and reported both a higher rate of Lp(a) > 30 mg/dL and a higher mean Lp(a) concentration in patients with type 1 and type 2 DM in comparison to healthy patients [23]. Although high levels of Lp(a) have been proven to increase the risk of atherosclerosis development among diabetic patients, Lp(a) concentration and DM occurrence appear to be inversely correlated. However, it remains unclear whether this association is causal [23,24].

Both elevated BMI and high Lp(a) levels are well-established CV risk factors. However, the evidence regarding the relationship between BMI and Lp(a) is heterogenous. Konieczyńska et al. reported similar median BMI values regardless of the Lp(a) group [18]. However, in all groups, the median BMI was greater than 25 kg/m^2^, suggesting there was an increased proportion of patients with overweight and obesity. In the large study by Kaltoft et al., the Lp(a) concertation increased slightly by 0.097 mg/dL, with an increase in BMI by 1 kg/m^2^ after adjustment for age and gender [25]. On the contrary, in a study conducted on a Chinese population, a weak but significant negative correlation between Lp(a) concentration and BMI (R = −0.0232) was found [26]. The relationship between these two CV risk factors requires further investigation.

The distribution of CV risk factors in both groups was similar to the normal distribution. However, in the high-Lp(a) group, the distribution was more diffused, with a less marked peak. The distribution of abnormal lipid parameters was also close to normal among all the Lp(a) groups. These results indirectly support the major role of genetic factors in determining the concentration of Lp(a), which have been proven to be associated with over 90% of the variance of Lp(a) plasma concentrations [27]. Non-genetic factors such as hormonal disbalance and kidney or liver diseases may modify the remaining part [13,28].

The prevalence of an elevated Lp(a) concentration ≥50 mg/dL varies from 6.0% to 24.0% in different populations [19,21,29]. In patients with diagnosed coronary artery disease, it might be even higher, ranging from 27.9% to 36.1% [30,31]. The measurement of Lp(a) in Poland is still rare in everyday clinical practice, and, therefore, the knowledge of its epidemiology is insufficient. Konieczyńska et al. reported an Lp(a) > 50.0 mg/dL in 18% of middle-aged patients [18]. The measurement of Lp(a) concentration enables screening for patients at high CV risk. Those with extremely high concentrations (≥180 mg/dL) are at particularly high risk of ASCVD, comparable to heterozygous familial hypercholesterolemia [16].

With the increasing evidence, an impact of high Lp(a) values on the development of CVD is indisputable. However, as supported by the presented results, the relationship between Lp(a) and the occurrence of conventional CV risk factors is rather weak. Therefore, there is a need for a wider assessment of Lp(a) concentrations in patients at risk of developing ASCVD in everyday clinical practice. This may help find patients with an additional CV risk and enable more watchful management. Furthermore, the assessment of Lp(a) might be beneficial for secondary prevention patients, who have low burdens of conventional CV risk factors [32]. To date, no potent treatment targeted at lowering Lp(a) is available [32]. However, some advanced clinical trials on several therapeutic options are being conducted. To balance the risk associated with an elevated Lp(a) concentration, it is recommended to adequately manage other known CV risk factors [33].

The major limitation of the presented analysis is the relatively small number of patients included in it. Nevertheless, this study shows the Lp(a) concentrations in one of the largest Polish cohorts of patients without ASCVD and their association with major CV risk factors. However, some factors, like kidney function or hormonal disbalance, that may influence Lp(a) concentrations were not assessed and require evaluation in the future.

## 5. Conclusions

Elevated Lp(a) concentration was not related to other impaired lipid parameters and the number of conventional CV risk factors. The prevalence of previously diagnosed hyperlipidemia was higher in patients with high Lp(a) concentrations and of an older age, and increased BMI values were less likely to occur. The occurrence of hypertension, DM, smoking status, and inadequate physical activity were not related to Lp(a) concentrations. The relationship between Lp(a) and sdLDL-C requires further investigation.

## Figures and Tables

**Figure 1 jcm-13-04649-f001:**
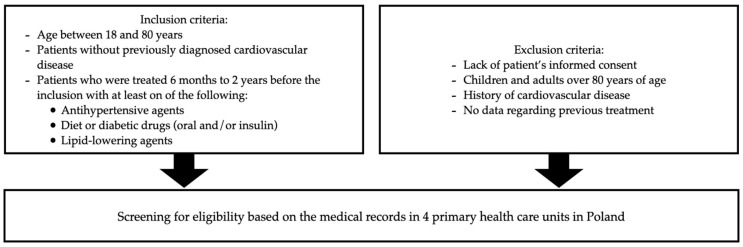
Inclusion and exclusion criteria.

**Figure 2 jcm-13-04649-f002:**
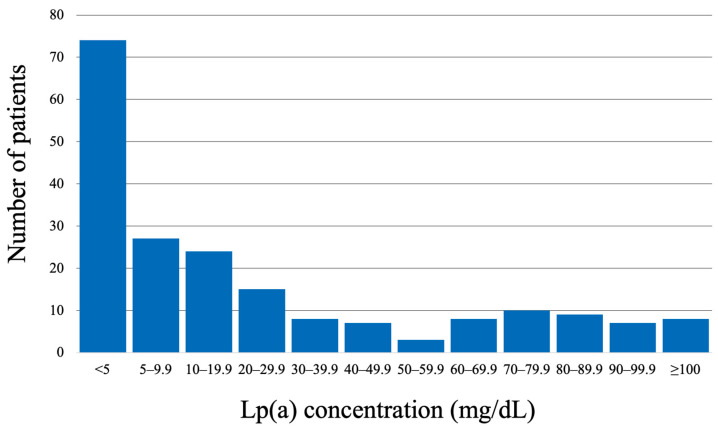
Distribution of Lp(a) concentrations for the studied population.

**Figure 3 jcm-13-04649-f003:**
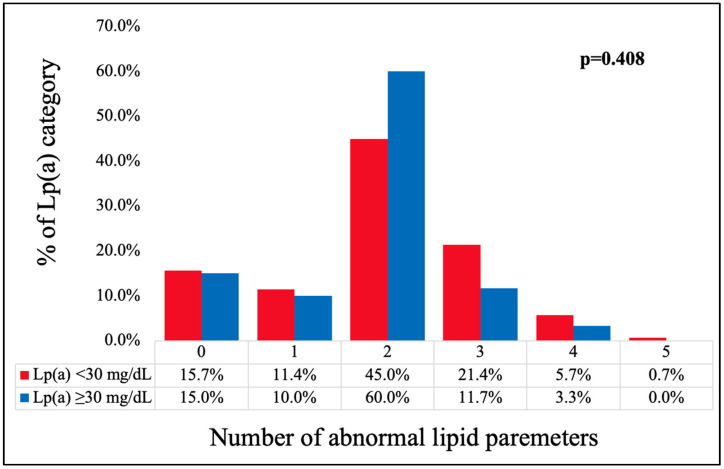
Distribution of the number of impaired lipid factors (TC > 4.9 mmol/L, non-HDL-C > 3.4 mmol/L, HDL-C < 1.0 mmol/L in men or <1.2 mmol/L in women, TG > 1.7 mmol/L, sdLDL > 1.0 mmol/L) according to the Lp(a) categories. [TC—total cholesterol; HDL-C—high-density lipoprotein cholesterol; non-HDL-C—non-high-density lipoprotein cholesterol; TG—triglycerides; sdLDL-C—small, dense LDL-C; Lp(a)—lipoprotein (a)].

**Figure 4 jcm-13-04649-f004:**
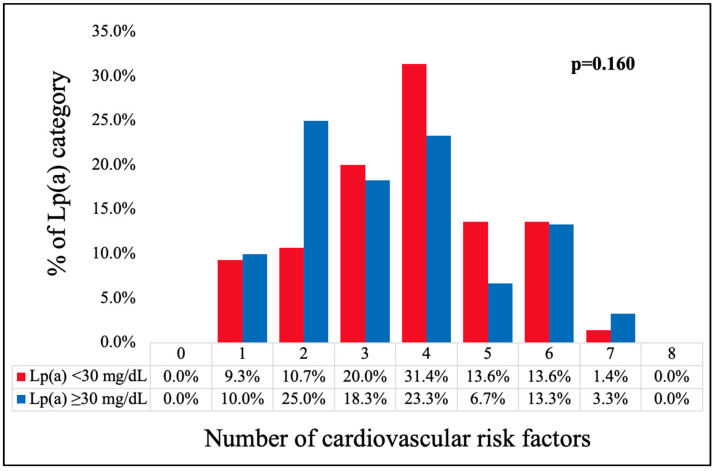
Distribution of the number of CV risk factors (age ≥ 60 years, history of hypertension, history of DM, history of hyperlipidemia, BMI ≥ 25 kg/m^2^, elevated AC ≥ 94 cm for men and ≥80 cm for women, smoker status, inadequate physical activity level) according to the Lp(a) categories. [DM—diabetes mellitus, BMI—body mass index, AC—abdominal circumference, Lp(a)—lipoprotein (a)].

**Table 1 jcm-13-04649-t001:** Baseline characteristics of the studied population.

Analyzed Parameter		Value
Age, median (IQR)		52.0 (43.0–60.0)
Higher education, *n* (%)		117 (58.5)
Male gender, *n* (%)		133 (66.5)
Marital status, *n* (%)	Low	30 (15.0)
	Average	141(70.5)
	High	29 (14.5)
History of hyperlipidemia, *n* (%)		105 (52.5)
Lipid-lowering drugs, *n* (%)		92 (46.0)
History of hypertension, *n* (%)		140 (70.0)
History of DM, *n* (%)		41 (20.5)
AC [cm], median (IQR)		87.0 (80.0–95.5)
AC, *n* (%)	Normal	74 (37.0)
	Overweight [F ≥ 80 cm and <88 cm, M ≥ 94 cm and <102 cm]	57 (28.5)
	Obesity [F ≥ 88 cm, M ≥ 102 cm]	69 (34.5)
BMI [kg/m^2^], median (IQR)		26.0 (24.0–28.7)
BMI category, *n* (%)	Underweight < 18.5 kg/m^2^	6 (3.0)
	Normal 18.5–24.9 kg/m^2^	77 (38.5)
	Overweight 25.0–29.9 kg/m^2^	84 (42.0)
	Obesity > 30.0 kg/m^2^	33 (16.5)
Smoker, *n* (%)		30 (15.0)
Inadequate physical activity, *n* (%)		119 (59.5)
TC [mmol/L], median (IQR)		5.56 (4.91–6.26)
non-HDL [mmol/L], median (IQR)		3.99 (3.39–4.68)
LDL-C [mmol/L], median (IQR)		3.29 (2.68–4.0)
HDL-C [mmol/L], median (IQR)		1.50 (1.25–1.81)
TG [mmol/L], median (IQR)		1.21 (0.90–1.55)
sdLDL, [mmol/L], median (IQR)		0.64 (0.53–0.78)
Lp(a) [mg/dL], median (IQR)		9.19 (3.54–42.07)
Lp(a) category, *n* (%)	Lp(a) < 30 mg/dL	140 (70.0)
	Lp(a) ≥ 30 mg/dL	60 (30.0)
FPG [mmol/L], median (IQR)		5.42 (5.04–5.88)
hsTnI [ng/L], median (IQR)		2.3 (1.6–3.2)
CRP [mg/L], median (IQR)		0.9 (0.6–1.7)
Serum creatinine [mg/dL], median (IQR)		0.8 (0.7–0.9)

Note: AC—abdominal circumference; BMI—body mass index; CRP—C-reactive protein; DM—diabetes mellitus; F—females; FPG—fasting plasma glucose; hsTnI—high-sensitivity cardiac troponin I; IQR—interquartile range; M—males; LDL-C—low-density lipoprotein cholesterol; Lp(a)—lipoprotein (a); sdLDL-C—small, dense LDL-C; TC—total cholesterol; TG—triglycerides.

**Table 2 jcm-13-04649-t002:** Lipid disorders and cardiovascular risk factors according to the Lp(a) categories.

CV Risk Factor	Total Studied Group, *n* (%)	Lp(a) <30 mg/dL, *n* (%)	Lp(a) ≥30 mg/dL, *n* (%)	*p*-Value
Age ≥ 60 years	55 (27.5)	46 (32.9)	9 (15.0)	0.01
History of hyperlipidemia	105 (52.5)	66 (47.1)	39 (65.0)	0.02
History of hypertension	140 (70.0)	101 (72.1)	39 (65.0)	0.312
History of DM	41 (20.5)	24 (17.1)	17 (28.3)	0.072
AC overweight and obesity	126 (63.0)	94 (67.1)	32 (53.3)	0.064
BMI overweight and obesity	117 (58.5)	89 (63.6)	28 (46.7)	0.026
Smoker	30 (15.0)	21 (15.0)	9 (15.0)	0.999
Inadequate physical activity	119 (59.5)	85 (60.7)	34 (56.7)	0.593
TC > 4.9 mmol/L (>190 mg/dL)	152 (76.0)	108 (77.1)	44 (73.3)	0.563
Non-HDL-C > 3.4 mmol/L (>130 mg/dL)	148 (74.0)	102 (72.9)	46 (76.7)	0.574
LDL-C > 3.0 mmol/L (>115 mg/dL)	122 (61.0)	83 (59.3)	39 (65.0)	0.448
HDL-C—M < 1.0 mmol/L (<40 mg/dL), F < 1.2 mmol/L (<45 mg/dL)	24 (12.0)	16 (11.4)	8 (13.3)	0.704
TG > 1.7 mmol/L (>150 mg/dL)	37 (18.5)	29 (20.7)	8 (13.3)	0.218
sdLDL > 1.0 mmol/L (>40 mg/dL)	15 (7.5)	14 (10.0)	1 (1.7)	0.04

Note: AC—abdominal circumference; BMI—body mass index; DM—diabetes mellitus; F—females; M—males; LDL-C—low-density lipoprotein cholesterol; Lp(a)—lipoprotein (a); sdLDL-C—small, dense LDL-C; TC—total cholesterol; TG—triglycerides.

**Table 3 jcm-13-04649-t003:** Concentrations of lipid parameters according to the Lp(a) categories.

Lipid Parameters	Lp(a) <30 mg/dL	Lp(a) ≥30 mg/dL	*p*-Value
TC [mmol/L], median (IQR)	5.56 (4.98–6.41)	5.55 (4.82–6.20)	0.537
non-HDL-C [mmol/L], median (IQR)	4.01 (3.35–4.76)	3.95 (3.43–4.57)	0.539
LDL-C [mmol/L], median (IQR)	3.29 (2.66–4.02)	3.24 (2.76–3.83)	0.904
HDL-C [mmol/L], median (IQR)	1.5 (1.24–1.83)	1.51 (1.26–1.77)	0.995
TG [mmol/L], median (IQR)	1.28 (0.92–1.59)	1.11 (0.87–1.40)	0.109
sdLDL, [mmol/L], median (IQR)	0.65 (0.53–0.81)	0.60 (0.52–0.72)	0.126

Note: IQR—interquartile range; TC—total cholesterol; HDL-C—high-density lipoprotein cholesterol; non-HDL-C—non-high-density lipoprotein cholesterol; TG—triglycerides; sdLDL-C—small, dense LDL-C; Lp(a)—lipoprotein (a).

## Data Availability

The data presented in this study are available from all the authors.

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
