# Peer review of "Lipoprotein (a) and the Occurrence of Lipid Disorders and Other Cardiovascular Risk Factors in Patients without Diagnosed Cardiovascular Disease"

_jcm, 2024, doi:10.3390/jcm13164649_

Round 1
Reviewer 1 Report
Comments and Suggestions for Authors
Ratajczak Jakub et al. present an article with the aim to assess the occurrence of lipid disorders and other CV risk factors according to the Lp(a). The article is interesting but key inputs need to be considered to improve the quality and scientific impact of the manuscript.
1. I suggest to update the references in the Introduction with latest guidelines and consensus on this topic and the dyslipidemia.
2. I suggest to include a figure with the study population and inclusion and exclusion criteria.
3. What is the impact of your findings in the clinical practice? Discuss this point in the Discussion. Lipoprotein(a) is a risk factor for atherosclerosis, how does it relate to your results and management of the patients?
Please, add this key reference to improve the scientific content of the new sentence: Di Fusco SA, et al. Lipoprotein(a): a risk factor for atherosclerosis and an emerging therapeutic target. Heart. 2022 Dec 13;109(1):18-25. doi: 10.1136/heartjnl-2021-320708. PMID: 35288443.
4. What are the future directions and gaps in evidence? Add a specific paragraph.
5. It is recommended to add a central figure to summarize the role of Lp(a) based on the results obtained.
Reviewer 2 Report
Comments and Suggestions for Authorsü provide an explanation for the abbreviation BMI in the abstract
ü line 37, correct the decimal notation, specify a period instead of a comma
ü at the beginning of the paragraph related to the methodology, state the type of study and the time frame of the realization of the study
ü separate the description of statistical methods in a separate paragraph
ü the authors state that continuous variables are reported as median with interquartile range. It should be emphasized that this way of presentation is used in case the variable does not have a normal distribution, otherwise it is necessary to state the mean value with the standard deviation
ü in the description of statistics, state which test was used to compare continuous variables (eg p values for two subgroups of respondents according to Lp(a), Table 3)
ü state the calculation for the power of the study, how the authors came to the conclusion that 200 respondents are sufficient
ü it is necessary to technically correct graph 1, so that the values of the x axis are placed horizontally, not diagonally
ü the quality of the manuscript would be improved if a regression analysis of the influence of independent variables on the occurrence of lipid disorders (cardiovascular risk) was performed
ü avoid repeating the results in the discussion, but discuss them from the perspective of previously conducted studies with a similar theme and design
Round 2
Reviewer 1 Report
Comments and Suggestions for Authors
The authors have answered my doubts and comments satisfactorily, congratulations
Reviewer 2 Report
Comments and Suggestions for Authors
The authors generally accepted the reviewer's suggestions, which improved the quality of the work. The topic of the manuscript is very current, bearing in mind the global epidemic of hyperlipoproteinemia and lipidosis. Given that the manuscript follows the latest national guidelines for disorders of lipid metabolism, it gives additional importance to the material.